# Redox Biology and Liver Fibrosis

**DOI:** 10.3390/ijms25010410

**Published:** 2023-12-28

**Authors:** Francesco Bellanti, Domenica Mangieri, Gianluigi Vendemiale

**Affiliations:** 1Department of Medical and Surgical Sciences, University of Foggia, 71122 Foggia, Italy; gianluigi.vendemiale@unifg.it; 2Department of Clinical and Experimental Medicine, University of Foggia, 71122 Foggia, Italy; domenica.mangieri@unifg.it

**Keywords:** redox homeostasis, stellate cell activation, chronic liver disease

## Abstract

Hepatic fibrosis is a complex process that develops in chronic liver diseases. Even though the initiation and progression of fibrosis rely on the underlying etiology, mutual mechanisms can be recognized and targeted for therapeutic purposes. Irrespective of the primary cause of liver disease, persistent damage to parenchymal cells triggers the overproduction of reactive species, with the consequent disruption of redox balance. Reactive species are important mediators for the homeostasis of both hepatocytes and non-parenchymal liver cells. Indeed, other than acting as cytotoxic agents, reactive species are able to modulate specific signaling pathways that may be relevant to hepatic fibrogenesis. After a brief introduction to redox biology and the mechanisms of fibrogenesis, this review aims to summarize the current evidence of the involvement of redox-dependent pathways in liver fibrosis and focuses on possible therapeutic targets.

## 1. Introduction

Persistent damage to parenchymal cells in chronic liver disease activates a network of signaling pathways, with transdifferentiation of hepatic stellate cells (HSCs) and portal fibroblasts toward myofibroblasts [1,2,3,4]. Activated myofibroblasts mostly produce type I and type III collagen, which crosslink and accumulate in the extracellular matrix (ECM) to replace injured liver parenchyma, resulting in a fibrous scar [5]. Thus, fibrosis in the liver can be considered as a healing reaction to various injury types, which include both hepatotoxic (chronic hepatocellular insult) and cholestatic (bile flow obstruction) damage [1]. HSCs represent the main source of collagen in chronic hepatocellular injury, while portal fibroblasts are determinant for fibrosis in chronic cholestatic conditions [3].

Inflammation associated with chronic liver diseases is critical for the initiation and progression of liver fibrosis. Indeed, fibrogenic liver injury is triggered and perpetuated by Kupffer cells (KCs), bone marrow-derived macrophages, and neutrophils, which release cytokines, chemokines, and growth factors with a critical role in the pathogenesis of fibrosis [6]. In particular, KCs and recruited macrophages produce not only profibrogenic cytokines, including transforming growth factor β (TGF-β) and platelet-derived growth factor (PDGF), but also several matrix metalloproteases (MMP) and tissue inhibitors of metalloproteases (TIMP) that regulate ECM turnover [7]. In this context, reactive compounds derived from alterations in redox homeostasis contribute to the initiation and progression of fibrosis in the liver. In almost all chronic hepatic conditions, including alcoholic and non-alcoholic liver disease, viral hepatitis, and autoimmune and cholestatic disorders, the overproduction of reactive species by liver parenchymal cells and immune cells, as well as endogenous antioxidant depletion, occurs [8]. The resulting changes in redox balance may directly activate transdifferentiation to fibrogenic cells in the liver, further amplifying the inflammatory network that promotes liver fibrosis [9]. Moreover, evidence that antioxidant compounds exert antifibrotic properties supports the strong interconnection between redox disbalance and inflammation in the evolution of chronic liver disease toward hepatic fibrosis [10].

After a brief overview of redox balance in the liver, the present review will highlight aspects related to redox homeostasis and liver fibrosis, focusing on altered redox-dependent pathways and ferroptosis. The main findings related to the use of molecules with antioxidant properties that are able to reverse liver fibrosis will also be discussed.

## 2. Redox Biology in the Liver

Carbohydrate, lipid, and ammonia metabolism, endogenous and exogenous toxic biotransformation, and bile synthesis occur in the liver. These different metabolic pathways can efficiently run in parallel because the liver parenchyma is specialized through metabolic zonation characterized by a specific oxygen supply to lower futile cycles [11]. Moreover, different hepatic zones are characterized by dynamic gene expression patterns and enzyme distribution, which are dependent not only on nutrients and hormones but also on concentrations of oxygen and reactive species [12]. The main regulators of metabolic reactions in parenchymal liver cells are the following:Subcellular organelles—most oxidative reactions occur in mitochondria and peroxisomes, while the cytosol is the main cellular site of reductive reactions [13];Disposal of coenzymes—oxidized/reduced NAD (NAD^+^/NADH) in oxidative (catabolic) reactions and oxidized/reduced NAD phosphate (NADP^+^/NADPH) in reductive (anabolic) reactions [14];Cellular AMP/ATP ratio—reduced ATP generation and/or higher ATP consumption trigger AMP-activated protein kinase (AMPK), promoting catabolism; on the other side, AMPK is inhibited by increased ATP disposal, boosting anabolism [15].

### 2.1. Hepatic Sources of Reactive Species

The term ‘reactive species’ refers to both reactive oxygen (ROS) and nitrogen (RNS) species in this review. Hepatic reactive species are generated by metabolic reactions occurring in several cell types and subcellular compartments (Figure 1). Potential inducers of reactive species include heavy metals, pollutants, smoke, xenobiotics, microplastics, drugs, or radiation.

Xanthine oxidoreductase (XOR) catalyzes the last two steps of purine catabolism in the cytosol of parenchymal liver cells, with the final oxidation of hypoxanthine to xanthine and then to uric acid [16]. Acting as a dehydrogenase in the human liver, XOR transfers electrons to NAD^+^; however, several stimuli can induce XOR to act as an oxidase, with electron transfer to O_2_ and the production of reactive species [17].

Mitochondrial respiratory chain complexes transfer electrons to NAD^+^, flavin mononucleotide, and flavin adenine dinucleotide (FAD), reducing O_2_ in a multiphasic pathway that generates reactive species [18]. In the liver, mitochondria physiologically produce 13–15% of H_2_O_2_ per 2% of O_2_ consumed [19]. Most reactive species in mitochondria are generated by Complexes I and III, although Complex II may act as a facultative producer [20,21]. Minor mitochondrial producers of reactive species are cytochrome b5 reductase [22] and monoamine oxidases A/B [23] in the outer membrane; pyruvate dehydrogenase and α-ketoglutarate dehydrogenase in the matrix [24]; dihydroorotate dehydrogenase [25], α-glycerophosphate dehydrogenase [26], ubiquinone oxidoreductase [27], proline dehydrogenase, and the branched-chain α-ketoacid dehydrogenase complex [24] in the inner membrane.

Peroxisomes are involved in fatty acid catabolism, the metabolism of pentose phosphates and D-amino acids, alternative alcohol oxidation, and NAD^+^ regeneration [28]. Moreover, XOR and the inducible isoform of nitrate synthase are located within peroxisomes [29,30]. These organelles are responsible for about 20% of O_2_ uptake and 35% of H_2_O_2_ generation, producing more reactive species than mitochondria in the liver [31].

The endoplasmic reticulum (ER) accounts for (1) the metabolism of lipids, steroids, and xenobiotics, (2) protein synthesis, folding, and trafficking, and (3) calcium storage [32]. The microsomal monooxygenase system, involved in lipid and steroid metabolism, is one of the main sources of reactive species in the ER. Xenobiotic metabolism occurs via phase I monooxygenation reactions (catalyzed by cytochrome P450 and the flavoprotein NADPH-cytochrome P450 reductase) and phase II conjugation reactions. Electron transfer from NADPH to P450 in phase I results in a leakage that produces reactive species [33]. Further leakage that produces reactive species is described in the electron transfer process, which is mediated by NADH-cytochrome b5 reductase in fatty acid desaturation [34]. Protein folding requires a high oxidized (GSSG)-to-reduced (GSH) glutathione ratio in the ER lumen to oxidize sulfhydryl groups [35]. In hepatocytes, enzymes involved in electron transfer for protein folding include ER oxidoreductin 1, which uses O_2_ as a final acceptor, producing ~25% reactive species [36]. 

Being essential for autophagy, lysosomes account for hepatocellular energy balance by regulating substrate availability, metabolic enzymes, and mitochondria quality [37]. Lysosomes counteract an excess of reactive species by removing damaged mitochondria, toxic cellular compounds, and unfolded proteins [38,39]. Lysosomes contain an electron transport chain, including ubiquinone, which is reduced by cytosolic NADH using O_2_ as the final acceptor; the acidification of the lysosomal matrix causes the partial reduction of O_2_, producing reactive species [40].

Reactive species can be further produced by hepatic NADPH oxidase (NOX) and NO synthases (NOS). NADPH oxidase is located in both parenchymal and nonparenchymal liver cells [41]. Phagocytic KCs contain NADPH oxidase (NOX2), which produces high quantities of reactive species [42]. HSCs further express a non-phagocytic NADPH oxidase isoform (NOX1), which generates mild amounts of reactive species [43,44,45]. Hepatic NO synthases are constitutive (eNOS) in liver sinusoidal endothelial cells and inducible (iNOS) in hepatocytes, KCs, HSCs, and liver sinusoidal endothelial cells. The production of NO by eNOS is determinant to maintain hepatic blood flow [46]. In the liver, iNOS expression can be modulated by several cytokines and is associated with the production of reactive species that can have both harmful and protective effects [46,47].

### 2.2. Hepatic Antioxidants

To preserve redox balance, reactive species are neutralized by several hepatic antioxidants. Classified as non-enzymatic and enzymatic, endogenous antioxidants interact via a complex network of redox reactions between subcellular compartments and the cytosol (Figure 2).

Non-enzymatic antioxidants include GSH, ubiquinone (UQ), and thioredoxin (TRX). GSH represents the most concentrated antioxidant in liver cells and is composed of three peptides with a sulfhydryl group in a cysteine residue that acts as a reductant on oxidized enzymes and antioxidants [48]. Liver cells contain lipophilic UQ (or coenzyme Q), mostly present in the membranes of Golgi vesicles, followed by mitochondrial and lysosomal membranes [49]. In its reduced form (ubiquinol, UQH2), coenzyme Q acts as an antioxidant. Nevertheless, its partial reduced form (ubisemiquinone, UQ•^−^) can cycle through three different redox states, allowing electron transfer activity in the mitochondrial respiratory chain from Complex I or II to Complex III [50]. The tetrapeptide TRX contains two sulfhydryl groups in two cysteine residues and is involved in reversible redox reactions catalyzed by NADPH-dependent thioredoxin reductase. In its reduced form, TRX reduces oxidized peroxiredoxin (PRX), which contributes mainly to the preservation of redox homeostasis in the liver [51].

Enzymatic antioxidants include superoxide dismutase (SOD), catalase (CAT), glutathione reductase (GR), thioredoxin reductases (TRXRs), glutathione peroxidase (GPX), and PRX. The liver expresses the highest quantity of SOD in humans [52]. SOD isoforms contain Cu/Zn (SOD1 and SOD3) or Mn (SOD2) in their active sites and are responsible for the dismutation of the superoxide anion [53]. SOD1 and SOD3 are located mainly in the cytosol of lysosomes, while SOD3 is mostly located in the mitochondria [54]. CAT is an iron-dependent peroxidase that converts two H_2_O_2_ into two H_2_O and one O_2_. In humans, CAT activity is highest in the liver and erythrocytes [55]. The hepatic GR and TRXRs (cytosolic TRXR1 and mitochondrial TRXR2) reduce disulfides to dithiols using NADPH: GR reduces GSSG, while TRXRs (as selenoproteins) form selenothiol pairs that reduce TRX [56]. GPX isoforms are selenium-dependent peroxidases that are oxidized via the conversion of H_2_O_2_ to H_2_O or the conversion of organic hydroperoxide (ROOH), which is reduced by GSH, to its corresponding alcohol (ROH). Among eight GPX isoforms described in humans, GPX1, GPX2, GPX4 (phospholipid hydroperoxidase), and GPX7 are located in the liver [57]. GPX1, GPX2, and GPX7 target H_2_O_2_ in the cytosol, mitochondria (GPX1), extracellular space (GPX2 and GPX7), and ER (GPX7), while GPX4 targets cytosolic, mitochondrial, and nuclear ROOH [57]. PRX (thiol hydrolases) can be oxidized by H_2_O_2_ or ROOH and reduced by TRX. The human liver expresses six PRX isoforms that target both H_2_O_2_ and ROOH in the cytosol (PRX1, PRX2, PRX5, and PRX6), mitochondria (PRX3 and PRX5), extracellular space (PRX4), nucleus (PRX5), and endosomes (PRX3 and PRX6) [58].

## 3. Redox-Dependent Mechanisms of Hepatic Fibrosis

Liver fibrosis develops from the uncontrolled accumulation of ECM as an end-stage manifestation of a scarred organ with an altered structure. Despite similar aspects, this process may follow hepatotoxic or cholestatic injury, dependent on the underlying cause of liver disease. Alcoholic and non-alcoholic liver disease, as well as chronic viral hepatitis, represent the main causes of hepatotoxic-induced liver fibrosis [59]. Cholestatic fibrosis may be the consequence of mechanical or immune-mediated damage to bile ducts (due to chronic pancreatitis or during primary biliary cholangitis/sclerosing cholangitis, respectively) or caused by congenital diseases (such as biliary atresia) [60].

Regardless of their etiology, both hepatotoxic and cholestatic liver diseases are characterized by an extremely oxidative environment that extends hepatocellular injury, supporting the development and progression of fibrosis [61]. Lipid peroxidation in hepatocytes, as well as neutrophil- and CYP2E1-derived reactive species, promotes type I collagen expression in HSCs [62,63,64]. H_2_O_2_ and IL-6 are able to facilitate the profibrogenic stimuli of KCs on HSCs [65]. The production of ROS by both NOX1 and NOX2 isoforms in HSCs and KCs exerts a profibrogenic effect [66]. The crosstalk between hepatic redox alterations and ER stress is also described as fibrogenic, since H_2_O_2_- or ethanol-induced UPR activates HSCs [67]. However, the effect of RNS seems controversial because nitric oxide is shown as protective against liver fibrosis [68], and NO-derived reactive species may downregulate fibrogenesis and prevent the activation of HSCs [69]. Nevertheless, the depletion of iNOS decreases fibrosis in a CCl_4_ model of liver injury [70], and peroxynitrite contributes to the activation of MMP2 secreted by HSCs and consequent ECM remodeling [71]. Most studies support the hypothesis that RNS may be antifibrogenic in the early phase of fibrosis development but have no or profibrogenic effects in late fibrogenesis (Figure 3) [9].

The direct involvement of redox biology in hepatic fibrogenesis is supported by evidence that reactive species and antioxidants may finely modulate key molecular mechanisms of liver fibrosis, such as the TGF-β, Wnt, and Hedgehog signaling pathways.

### 3.1. Redox Homeostasis and TGF-β Signaling Pathway in Liver Fibrosis

As a central modulator of hepatic fibrosis, the TGF-β pathway induces fibrogenesis via canonical (Smad-dependent) and non-canonical (Smad-independent) signals [72]. In the canonical pathway, TGF-β specifically binds to transmembrane receptors, promoting the phosphorylation of Smad2 and Smad3, which in turn form a heteromeric complex with Smad4 that translocates to the nucleus to induce the transcription of profibrogenic genes [73]. In the non-canonical pathway, TGF-β directly regulates Wnt/β-catenin, mTOR, phosphatidylinositol-3-kinase, MAP kinases, IKK, and Rho-like GTPase [74]. The secretion of TGF-β occurs as a huge latent complex, including dimeric TGF-β bound to latency associated protein (LAP) and latent TGF-β binding protein (LTBP) [75].

TGF-β can bind to specific receptors only after its release from LAP, which occurs through several mechanisms, including the direct oxidation of LAP or the indirect oxidation-dependent activation of MMPs that, in turn, cleave LAP [76,77]. Reactive species are also able to upregulate TGF-β mRNA expression through nuclear factor kappa-light-chain-enhancer of activated B cells (NF-κB) in liver cells infected by the hepatitis C virus [78]. Besides regulating TGF-β expression and activity, reactive species modulate several of its profibrotic effects. Indeed, mitochondria- or NADPH-derived reactive species are essential in the transdifferentiation of myofibroblasts induced by TGF-β [79,80]. A further mechanism of hepatic fibrosis is autophagy, which may be activated by the TGF-β-mediated production of reactive species in a NOX4-dependent pathway [81,82].

### 3.2. Redox Control of Wnt Signaling Pathway

Wnt signaling is crucial for liver fibrosis and HSC activation. The canonical and non-canonical Wnt-dependent pathways are described and rely on the involvement of β-catenin [83]. In the canonical pathway, β-catenin translocates from the membrane to the cytoplasm of injured hepatocytes, but it goes through proteasomal degradation if Wnt is inhibited; in contrast, activated Wnt leads to β-catenin translocation to the nucleus and the consequent transcription of target genes [84]. Non-canonical pathways are independent of β-catenin and activate gene transcription via an increased concentration of Ca^2+^ or through planar cell polarity and the consequent activation of Rac and Roc [85].

The Wnt pathway is regulated by reactive species through nucleoredoxin, a TRX-related protein, which interacts with Dishevelled (a fundamental adaptor protein for Wnt signaling), thereby inhibiting its activation [86]. An oxidative environment induces the dissociation of nucleoredoxin from Dishevelled, which activates Wnt signaling [86]. Even though mechanistic studies that define the redox-dependent modulation of the Wnt pathway specifically in liver fibrosis are lacking, a preclinical investigation showed that the bioflavonoid morin is able to suppress hepatic fibrogenesis through the inhibition of Wnt, which is associated with an antioxidant effect [87].

### 3.3. Redox Homeostasis and Hedgehog Signaling

Hedgehog signaling is a further regulator of liver fibrosis progression. This signaling relies on Hedgehog ligands, which are produced as precursor proteins that undergo cleavage and subsequent modification. Hedgehog proteins bind to the canonical receptor Patched (PTCH1), which in turn activates GPCR-like protein Smoothened (SMO), with consequent downstream signal transduction [88]. The production of Hedgehog ligands and the accumulation of Hedgehog-responsive cells in the liver is related to the extension of hepatic damage and fibrosis [89,90].

In a model of liver fibrosis triggered by hexavalent chromium exposure, reduced antioxidant activity and altered hepatic redox balance were matched with HSC activation and increased activity of the Hedgehog signaling pathway [91]. No more studies on the redox modulation of Hedgehog signaling in liver fibrosis are available, but evidence of the role exerted by reactive species on this pathway is provided in different settings. Indeed, redox signaling is critical in modulating the activity of the Sonic Hedgehog pathway and inducing downstream events in neurodegenerative conditions [92]. Increased cellular levels of NADH caused by the inhibition of mitochondrial glycerophosphate dehydrogenase (a component of the glycerophosphate shuttle) are able to block Hedgehog transcriptional output in a model of medulloblastoma [93]. Furthermore, the reciprocal regulation of H_2_O_2_ and Sonic Hedgehog is a determinant of embryonic development and regenerative processes [94,95].

## 4. Ferroptosis: A Further Link between Redox Homeostasis and Liver Fibrosis

In contrast to other classes of cell death, such as apoptosis, pyroptosis, and autophagy, ferroptosis is iron-dependent cell death characterized by large amounts of lipid peroxidation [96]. Further redox cellular pathways, including selenium-dependent GPX4 and ferroptosis suppressor protein-1/coenzyme Q axis, are involved in ferroptosis [97]. Ferroptosis is the main contributor in the pathogenesis of several hepatic diseases, such as alcoholic and nonalcoholic steatohepatitis, viral hepatitis, and hepatocellular carcinoma [98].

Altered iron homeostasis resulting from an increased uptake and a reduced storage of iron in the liver triggers the Fenton reaction and enzymatic oxygenation, with the consequent production of reactive species and lipid peroxidation, which are both considered hallmark features of ferroptosis [99]. Nevertheless, the process of lipid peroxidation and the consequent induction of ferroptosis mostly relies on crosstalk signaling from different cellular organelles, including mitochondria, peroxisomes, and lysosomes [100]. Cells undergoing ferroptosis have smaller mitochondria with condensed cristae and disrupted membranes, while no alterations occur in the nucleus [101].

Iron abundance is considered a predisposing factor to the development of liver fibrosis, as suggested by the pathogenic significance of ferroptosis in models of hepatic iron overload [102]. Ferroptosis may induce liver fibrosis, as suggested by models of the hepatocyte-specific knockout of the transferrin gene in the setting of a high-iron diet; this process can be prevented by ferrostatin-1, a ferroptosis inhibitor [103]. However, triggering ferroptosis in HSCs reduces liver fibrosis, such that compounds that are able to promote this mechanism are deemed antifibrotic [104,105]. Even though the role of ferroptosis in hepatic fibrogenesis warrants further investigation, current evidence suggests that its effect on liver fibrosis is cell-specific since it can be profibrogenic in hepatocytes but antifibrogenic in HSCs. Thus, the best therapeutic strategy would consist of activating ferroptosis selectively in HSCs, thereby saving hepatocytes.

## 5. Targeting Redox Homeostasis to Treat Liver Fibrosis

The primary targets of antifibrotic therapy include the ECM, activated HSCs, and myofibroblasts; nevertheless, indirect targets may be represented by different cell types and pathways (such as modulators of inflammation and/or the immune response) [106]. To date, several compounds have been identified, tested, and shown to exert antifibrotic effects in preclinical settings, but only a few of them progressed to clinical evaluation [107]. This is also true for compounds able to modify the hepatic redox environment, which demonstrated beneficial effects in animal models of liver fibrosis [108,109,110], but no effectiveness was proven in humans [8,111,112]. Thus, there is no current approved antifibrotic drug, so the only existing treatment choices are either the elimination of causal agents or hepatic transplantation in advanced cirrhosis [113]. The main limitation of antifibrotic compounds most probably relies on targeting a single mechanism or pathway. Thus, multitarget compounds or combination therapies need to be improved to produce beneficial effects. 

It is conceivable that the failure of common antioxidant compounds to revert chronic liver fibrosis may stem from the complexity of the mechanisms and interactions that regulate redox homeostasis [114]. As a consequence, redox balance should be targeted using molecules that are able to selectively modulate oxidant sources and/or reducing pathways (Table 1). Of note, some of these molecules are currently under investigation for the treatment of liver fibrosis both as single and combination therapies [115].

Some molecules target mitochondria to treat chronic liver fibrosis. The antioxidant compound ubiquinone was able to lower oxidative stress and improve hepatic fibrosis in rodents [116,117]. Mitoquinone (MitoQ, a mitochondria-targeted UQ) was able to limit liver fibrosis induced by CCl_4_ through JNK/YAP signaling [118]. Furthermore, MitoQ attenuated liver fibrosis by inhibiting HSC activation and enhancing mitophagy, which results in the selective removal of damaged mitochondria [119]. Mitophagy was also enhanced by melatonin, with the consequent amelioration of liver fibrosis in mice treated with CCl_4_ [120].

Innovative therapeutic strategies for liver fibrosis target redox enzymes (such as NOX isoforms) or redox-dependent nuclear factors (such as nuclear factor erythroid 2-related factor 2, NRF2, and peroxisome proliferator-activated receptor-γ, PPAR-γ). Setanaxib (GKT137831) is a dual NOX1/4 inhibitor currently undergoing clinical studies that demonstrated antifibrotic properties linked to redox modulation in several organs during preclinical investigations [121]. With regard to liver fibrosis, setanaxib was protective in models of nonalcoholic steatohepatitis and CCl_4_ treatment, as well as cholestasis induced via bile duct ligation, which also reduced oxidative stress [122,123]. In particular, setanaxib was able to inhibit profibrogenic genes both in primary human and rodent HSCs through a reduction in reactive species [124]. Selective activation of NRF2—master regulator of redox homeostasis—in hepatocytes reduced fibrosis in a mouse model of steatohepatitis [125]. Fibrosis amelioration and a lower expression of profibrogenic genes were also found in the livers of mice with steatohepatitis treated with acetylenic tricyclic bis(cyano enone), a strong NRF2 activator [126]. Another NRF2 activator, S217879, was reported to be effective in impairing the progression of liver fibrosis in experimental nonalcoholic steatohepatitis [127]. The use of piperine, which promotes NRF2 activation and nuclear translocation, reduced HSC activation, collagen deposition, and liver fibrosis in rodents through the inhibition of the TGF-β/SMAD signaling [128]. The nuclear factor PPAR-γ is able to inhibit the polarization of macrophages toward the proinflammatory phenotype and revert HSC activation through the modulation of redox homeostasis [129,130]. The beneficial effects of PPAR-γ on redox balance and liver fibrosis are mediated by its binding to the PPAR-γ response element on the NRF2 promoter and also by differentially modulating iNOS and eNOS activities [131].

**Table 1 ijms-25-00410-t001:** Redox-targeted compounds that demonstrated antifibrotic effects in preclinical settings.

Molecule and Formulation	Dosage	Targeted Mechanisms Associated with Fibrosis Reduction	Preclinical Model of Fibrosis	Reference No.
Solubilized ubiquinone (Coenzyme Q10)	10 and 30 mg/kg	Inhibition of TGF-β1 and alpha-SMA; upregulation of GCL and GSTA2 via NRF2	DMN-induced liver fibrosis in mice; H4IIE and MEF cells	[116]
Dietary coenzyme Q10 supplementation	1 mg/kg	Reduction in lipid peroxidation (4-HNE) and inflammation (IL-6 and TNF)	Maternal protein restriction and accelerated postnatal growth in rats	[117]
Intraperitoneal mitoquinone mesylate (mitoQ)	2 mg/kg	Inhibition of TGF-β and type I collagen; reduction in mitochondrial damage and ROS production; inhibition of JNK phosphorylation and YAP nuclear translocation	CCl_4_ by oral gavage for 8 weeks in mice	[118]
Oral melatonin	2.5, 5, and 10 mg/kg	Attenuation of mitochondrial swelling and improvement of mitophagy/mitochondrial biogenesis/dynamics	Intraperitoneal CCl_4_ for 8 weeks in rats	[120]
Oral (gavage) GKT137831 (setanaxib)	60 mg/kg	NOX4/NOX1 inhibition, reduction in ROS production and hepatocellular apoptosis	BDL in rats and mice	[122]
Oral (gavage) GKT137831 (setanaxib)	60 mg/kg	NOX4 inhibition, reduction in inflammation and increase in insulin sensitivity	Fast food diet in mice	[123]
GKT137831 (setanaxib)	20 μM	Suppression of ROS production and inflammatory and proliferative genes	Primary mouse HSCs treated with LPS, PDGF, or Shh	[124]
Oral (gavage) TBE-31	5 nmol/g	NRF2 activation, increase in fatty acid oxidation and lipoprotein assembly, decrease in ER stress, inflammation, and apoptosis	High-fat plus fructose diet for 16 or 30 weeks in mice	[126]
Oral (gavage) S217879	3 or 30 mg/kg	NRF2 activation, inhibition of de novo lipogenesis and proinflammatory genes	Methionine- and choline-deficient or AMLN diet for 4 weeks in mice	[127]
Piperine		NRF2 activation, inhibition of TGF-β1/Smad axis	CCl_4_ treatment in mice, AML-12 and LX-2 cells	[128]

Abbreviations: TGF-β1, transforming growth factor-β1; SMA, smooth muscle actin; GCL, glutamate-cysteine ligase; GSTA2, glutathione S-transferase A2; NRF2, nuclear factor erythroid 2-related factor 2; DMN, dimethylnitrosamine; MEF, mouse embryonic fibroblast; 4-HNE, 4-hydroxynonenal; IL-6, interleukin-6, TNF, tumor necrosis factor; ROS, reactive oxygen species; JNK, c-Jun N-terminal kinase; YAP, yes-associated protein; NOX, NADPH oxidase; BDL, bile duct ligation; HSCs, hepatic stellate cells; LPS, lipopolysaccharide; PDGF, platelet-derived growth factor; Shh, sonic hedgehog; TBE-31, acetylenic tricyclic bis(cyano enone); ER, endoplasmic reticulum; AMLN, amylin modified NASH; AML-12, alpha mouse liver-12.

## 6. Conclusions

Liver fibrosis is the product of the interaction between both hepatic and recruited cells in a complex network orchestrated by several mediators, including the products of redox metabolism and redox-dependent signaling. To date, only the inhibition of causal factors was able to revert experimental and human liver fibrosis, with the consequent restoration of redox balance. Nevertheless, there is a pressing need for effective antifibrotic therapies to decrease the global burden of chronic liver diseases. In an attempt to target redox balance for the treatment of liver fibrosis, the success of antioxidant compounds in preclinical models was not confirmed in clinical studies. Consequently, new cell-specific and selective redox modulators should be tested—either independently or in combination therapies—to demonstrate their effectiveness in reducing human liver fibrosis. 

## Figures and Tables

**Figure 1 ijms-25-00410-f001:**
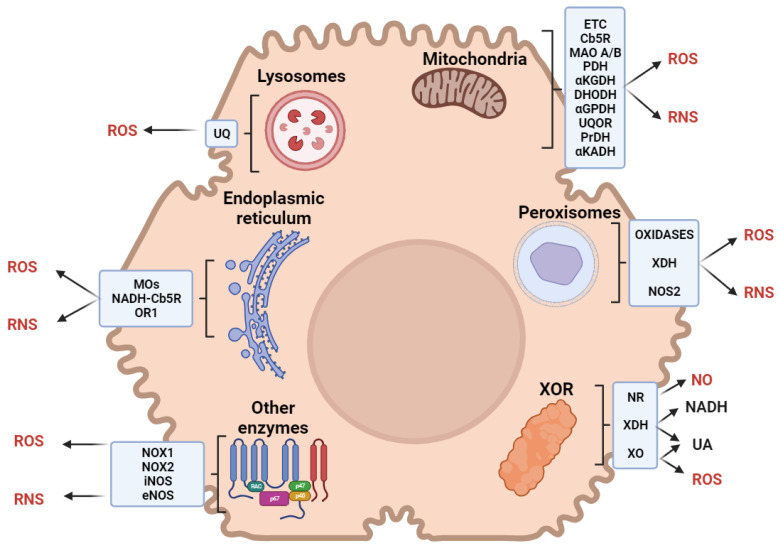
Sources of reactive oxygen (ROS) and nitrogen (RNS) species in liver cells. Abbreviations: ETC, electron transport chain; UQ, ubiquinone; Cb5R, cytochrome b5 reductase; MAO, monoaminoxidase; PDH, pyruvate dehydrogenase; αKGDH, α-ketoglutarate dehydrogenase; DHODH, dihydroorotate dehydrogenase; αGPDH, α-glycerophosphate dehydrogenase; UQOR, ubiquinone oxidoreductase; PrDH, proline dehydrogenase; αKADH, α-ketoacid dehydrogenase; MOs, monooxygenase; OR1, oxidoreductin 1; XOR, xanthine oxidoreductase; XDH, xanthine dehydrogenase; NOX, NADPH oxidase; NOS, nitric oxide synthase; NR, nitric oxide reductase; XO, xanthine oxidase.

**Figure 2 ijms-25-00410-f002:**
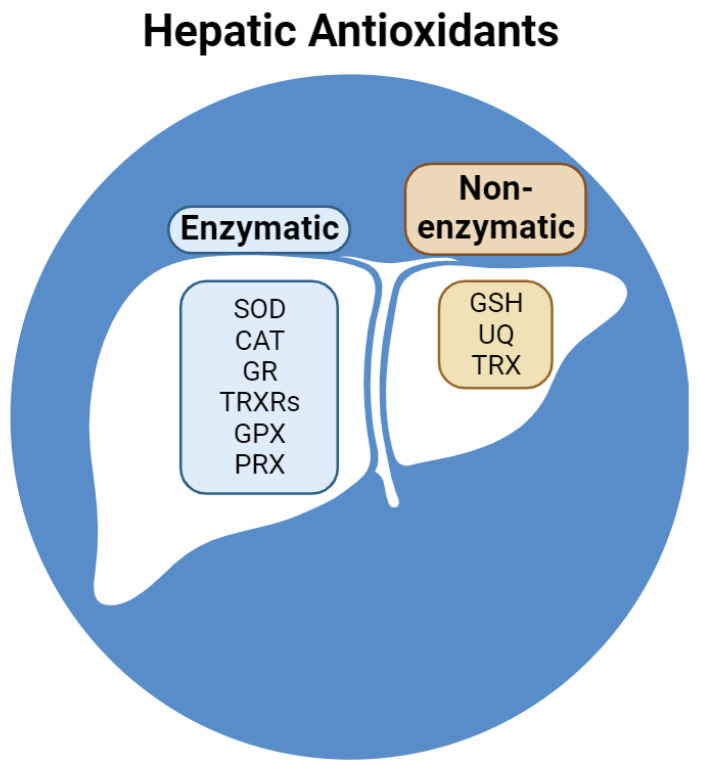
Reducing compounds and enzymes in the liver. Abbreviations: SOD, superoxide dismutase; CAT, catalase; GR, glutathione reductase; TRXRs, thioredoxin reductases; GPX, glutathione peroxidase; PRX, peroxiredoxin; GSH, reduced glutathione; UQ, ubiquinone; TRX, thioredoxin.

**Figure 3 ijms-25-00410-f003:**
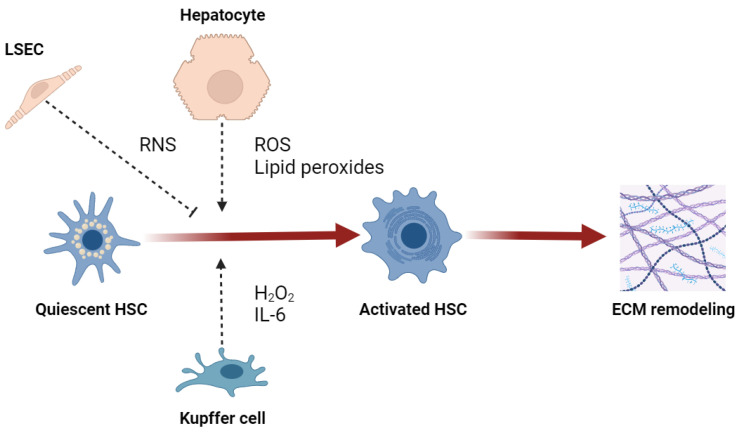
Effect of reactive species on activation of hepatic stellate cells (HSC). Abbreviations: LSEC, liver sinusoidal endothelial cell; ROS, reactive oxygen species; RNS, reactive nitrogen species; IL-6, interleukin-6; ECM, extracellular matrix.

## Data Availability

Not applicable.

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
