# Peer review of "Redox Biology and Liver Fibrosis"

_ijms, 2023, doi:10.3390/ijms25010410_

Round 1

Reviewer 1 Report

Comments and Suggestions for Authors

The review manuscript “Redox Biology and Liver Fibrosis” carefully explains redox-reactions and the reactants involved in these reactions in the liver. The text contains well-chosen references and the manuscript is of general interest. The concerns with the manuscript are all related to language. Please go through the text and edit the language, with specific focus on the following:

Line 239: “A canonical and non-canonical Wnt-dependent pathways are described”. Please correct to pathway.

Line 257: “This signaling relies on Hedgehog ligands, which are produced as precursor proteins that undergo cleavage, modification.” Do you mean cleavage and modification or maybe cleavage or modification? Please specify.

Line 268: “Indeed, redox signaling is critical in modulating the activity of Sonic Hedgehog pathway”. The Sonic Hedgehog pathway.

Line 285 “triggers Fenton reaction”, please change to triggers the Fenton reaction or triggers Fenton reactions

Line 295: “knock-out models of transferrin gene fed a high-iron diet”, please change to knock-out models of the transferrin gene

Line 334: “inhibiting HSCs activation by enhancing mitophagy”, please change to HSC, without the s.

Comments on the Quality of English Language

The language needs moderate editing. It is reasonably good, but there are some mistakes.

Author Response

The review manuscript “Redox Biology and Liver Fibrosis” carefully explains redox-reactions and the reactants involved in these reactions in the liver. The text contains well-chosen references and the manuscript is of general interest. The concerns with the manuscript are all related to language. Please go through the text and edit the language, with specific focus on the following:

Line 239: “A canonical and non-canonical Wnt-dependent pathways are described”. Please correct to pathway.

Line 257: “This signaling relies on Hedgehog ligands, which are produced as precursor proteins that undergo cleavage, modification.” Do you mean cleavage and modification or maybe cleavage or modification? Please specify.

Line 268: “Indeed, redox signaling is critical in modulating the activity of Sonic Hedgehog pathway”. The Sonic Hedgehog pathway.

Line 285 “triggers Fenton reaction”, please change to triggers the Fenton reaction or triggers Fenton reactions

Line 295: “knock-out models of transferrin gene fed a high-iron diet”, please change to knock-out models of the transferrin gene

Line 334: “inhibiting HSCs activation by enhancing mitophagy”, please change to HSC, without the s.

Reply: we thank the reviewer for his positive comments and careful reading. We checked the English language and modified all the suggested points.

Reviewer 2 Report

Comments and Suggestions for Authors

Thank you for submitting the article to International Journal of Molecular Sciences. Overall, the authors addressed well about the redox mechanisms, redox-associated liver fibrosis, and therapeutic targets. Here are several minor comments. 

Minor

  1. In 2.1 Hepatic sources of reactive species, the authors explained well how ROS and RNS could be induced in subcellular organelles. It is well described. But I want to suggest that it would be great to mention the representative inducers (e.g. chemicals, drugs).
  2. Here, the authors addressed the beneficial effects of drugs. However, it would be more helpful to let readers know the limitation and how we could improve these drugs. 

Comments on the Quality of English Language

Minor editing of English language required.

Author Response

Thank you for submitting the article to International Journal of Molecular Sciences. Overall, the authors addressed well about the redox mechanisms, redox-associated liver fibrosis, and therapeutic targets. Here are several minor comments. 

Reply: we thank the reviewer for his positive comments.

Minor

  1. In 2.1 Hepatic sources of reactive species, the authors explained well how ROS and RNS could be induced in subcellular organelles. It is well described. But I want to suggest that it would be great to mention the representative inducers (e.g. chemicals, drugs).

Reply: accordingly, we mentioned the representative inducers of reactive species (lines 76-78).

  1. Here, the authors addressed the beneficial effects of drugs. However, it would be more helpful to let readers know the limitation and how we could improve these drugs. 

Reply: to comply with the reviewer’s request, we addressed the main limitation of antifibrotic therapy adding a possible solution to improve its effect (lines 315-317).